# A WEIGHT VARIATION-AWARE TRAINING METHOD FOR HARDWARE NEUROMORPHIC CHIPS

## ABSTRACT

Hardware neuromorphic chips that mimic the biological nervous systems have recently attracted significant attention due to their ultra-low power and parallel computation. However, the inherent variability of nano-scale synaptic devices causes a weight perturbation and performance drop of neural networks. This paper proposes a training method to find weight with robustness to intrinsic device variability. A stochastic weight characteristic incurred by device inherent variability is considered during training. We investigate the impact of weight variation on both Spiking Neural Network (SNN) and standard Artificial Neural Network (ANN) with different architectures including fully connected, CNN, VGG, ResNet, MobileNet and EfficientNet on MNIST, N-MNIST, CIFAR-10, CIFAR-100, and ImageNet. Experimental results show that a *weight variation-aware training method (WVAT)* can dramatically minimize the performance drop on weight variability by exploring a flat loss landscape. On ferroelectric tunnel junctions (FTJ) devices, WVAT yields 78.01% accuracy of VGG-5 on CIFAR-10 for weight perturbations, while SGD scores 28.43%. Finally, WVAT is easy to implement on various architectures with little computational overhead.

## 1 INTRODUCTION

Deep Neural Networks (DNN) have achieved remarkable breakthroughs in computer vision, automatic driving, and image/voice recognition (LeCun et al., 2015). With this success, neuromorphic technology, which mimics the human nervous system, has recently received significant attention in the semiconductor industry. Compared with the conventional von Neumann architecture which has limitations in power consumption and real-time pattern recognition (Schuman et al., 2017; Indiveri et al., 2015), neuromorphic chips, biologically inspired from the human brain, are new compact semiconductor chips that collocate processing and memory (Chicca et al., 2014; Catherine D. Schuman & Kay, 2022). Therefore, neuromorphic chips can process highly parallel operations and be suitable for real-time recognizing images, videos, and audios with ultra-low power consumption (Indiveri & Liu, 2015).

Neuromorphic chips are also suitable for "Edge AI computing," which process data in edge devices rather than in the cloud at a data center (Nwakanma et al., 2021). In other words, tasks that require a large amount of computation, such as training, are performed in the cloud and inference in edge devices. Traditional cloud AI processing requires sufficient computing power and network connectivity. This means that an enormous amount of data transmission is required, likely increasing data latency and transferring disconnections (Li et al., 2020). It causes severe problems in autonomous driving, robotics, and mobile VR/AR that require real-time processing. Therefore, there is a growing need for data processing on edge devices. Neuromorphic devices are compact, mobile, and energy-efficient, promising candidates for edge computing systems.

However, despite enormous advances in semiconductor integrated circuit (IC) technology, hardware neuromorphic implementation and embedded systems with numerous synaptic devices remain challenging (Prezioso et al., 2015; Esser et al., 2015; Catherine D. Schuman & Kay, 2022). Design considerations such as multi-level state, device variability, programming energy, speed, and array-level connectivity, are required. (Eryilmaz et al., 2015). In particular, nano-electronic device variability is an inevitable issue originating from manufacturing fabrication (Prezioso et al., 2010). Although there are many kinds of nano-electronic devices for neuromorphic systems and in-memory computing–including memristor, flash memory, phase-change memory, ferroelectric devices, and optoelectronic devices–we call them "devices" for readability in this paper.

Device variability causes mapped synaptic weight values in hardware to differ undesirably from software weight, especially on analog synapses or neurons. This gap between hardware synapse and software weight makes it challenging to implement neural networks in real-world applications. Many recent studies have reported that device variability can significantly reduce the accuracy of neuromorphic hardware and DNN accelerators (Catherine D. Schuman & Kay, 2022; Peng et al., 2020; Joshi et al., 2020; Sun & Yu, 2019; Kim et al., 2019; 2018). Although there are various studies to solve this problem, they focus on the unique behaviors of devices (Hennen et al., 2022; vls; Fu et al., 2022). The diversity of devices used to implement neuromorphic hardware results in the customized solutions required for a given device variation. Therefore, the versatility of customized solutions at the device level is limited.

There is a growing need for a hardware-oriented training method to learn parameters robust to device variability. It is widely known that *wide and flat loss landscapes* lead to improved generalization (Keskar et al., 2017; Li et al., 2018). It is natural to think that wide and flat loss landscapes with respect to weight will mitigate the accuracy drop caused by device variability. However, we experimentally confirm that related studies (Izmailov et al., 2018; Wu et al., 2020; Foret et al., 2021) can not significantly reduce the accuracy drop by device variation (Experiments are provided later in section 2). This observation reminds us of the need for a hardware-oriented neural network training method.

Motivated by this, we propose a *weight variation-aware training method (WVAT)* that alleviates performance drops induced by device variability at the algorithmic level rather than the device level. This method explores a wide and flat *weight loss landscape* through the ensemble technique and the hardware-simulated variation-aware update method, which is more tolerant to the software weight perturbation caused by hardware synaptic variability. WVAT can effectively minimize performance drops with respect to weight variations with little additional computational overhead in the training phase. Our contributions include the following:

- For the first time to the best of our knowledge, we investigate and analyze the impact of variations in model parameters on performance in several architectures, including standard Artificial Neural Networks (ANN) and Spiking Neural Networks (SNN), which are suitable for hardware neuromorphic implementation due to event-driven spike properties.

- By exploring the flatter weight loss landscape, we propose WVAT that is tolerant to intrinsic device variability. We introduce an ensemble technique for better generalization and present a intuitive weight update method with a hardware-simulated variation. This method is also efficient for quantization and input noise, which is one of the hardware implementation issues besides weight perturbations.

- We experimentally demonstrate that WVAT achieves nearly similar performance to the typical training method stochastic gradient descent (SGD) while having robustness to variations in model parameters. WVAT is easy to implement with little computational cost.[1]

## 2 BACKGROUND

Many studies have been conducted to develop training methods robust to device variability (Liu et al., 2015; Long et al., 2019; Zhu et al., 2020; Joshi et al., 2020; Joksas et al., 2022; Huang et al., 2022). Liu et al. (2015) proposed adding a penalty for variations in model parameters to training loss. Long et al. (2019) and Zhu et al. (2020) generated a noise model to reflect device variability during a training phase. Although Long et al. (2019) achieved good performance, this method has a limitation, a binary device (1 bit per cell). However, as mentioned in section 1, the customized solutions for the given device has limitation in applying to the general case. (A comparison of WVAT with these methods is in Appendix A.1)

Recently, there have been many studies investigating the effect of loss landscape on generalization (Garipov et al., 2018; Izmailov et al., 2018; Wu et al., 2020; Foret et al., 2021; Liu et al., 2022). It is widely known that a flat loss landscape reduces the generalization gap. Stochastic weight averaging (SWA) (Izmailov et al., 2018) is an ensemble technique that averages the weights as a time axis instead of storing multiple models with different weights. It has been experimentally demonstrated that the ensemble method brings a flatter loss landscape, resulting in a lower test error than SGD.

---

[1] A source code will be available soon.

Adversarial weight perturbation (AWP) (Wu et al., 2020) and sharpness-aware minimization (SAM) (Foret et al., 2021) were proposed to improve model generalization by seeking the flat loss landscape. These methods explore the direction of weight perturbation in the worst case and update the weight based on that direction.

**AWP** AWP proposed a adversarial weight perturbation $\boldsymbol{v}$ based on the generated adversarial examples $x'_i$ for adversarial training:

$$\boldsymbol{v} = \frac{\nabla_{\boldsymbol{v}} \frac{1}{m} \sum_{i=1}^{m} l(f_{\boldsymbol{w}+\boldsymbol{v}}(x'_i), y_i)}{||\nabla_{\boldsymbol{v}} \frac{1}{m} \sum_{i=1}^{m} l(f_{\boldsymbol{w}+\boldsymbol{v}}(x'_i), y_i)||} ||\boldsymbol{w}||$$

where $y_i$ is label, and $m$ is batch size. $f_{\boldsymbol{w}}(\cdot)$ is neural network with weight $\boldsymbol{w}$, and $l$ is standard classification loss.

**SAM** SAM seeks model parameters whose entire neighborhoods have uniformly low training loss value. Neighborhood $\epsilon(w)$—maximizing loss value— is given by the solution of a classical dual norm problem:

$$\epsilon(\boldsymbol{w}) = \rho \, \text{sign}(\nabla_{\boldsymbol{w}} L_S(\boldsymbol{w})) \frac{|\nabla_{\boldsymbol{w}} L_S(\boldsymbol{w})|^{q-1}}{(||\nabla_{\boldsymbol{w}} L_S(\boldsymbol{w})||_q^q)^{\frac{1}{p}}}$$

where $1/p + 1/q = 1$, $\rho$ is neighborhood size, and $L_S(\cdot)$ is training set loss. In the case of $p = 2$, $\epsilon(\boldsymbol{w})$ is a norm of the gradient scaled by $\rho$.

Both methods have similarities in finding weight perturbations in the gradient-ascent direction, except the perturbation is scaled by a norm of weight (AWP) or neighborhood size (SAM). Although both methods yielded state-of-the-art performance, they have drawbacks in terms of computational overhead . An update rule in AWP and SAM requires two sequential gradient computations, one for obtaining weight perturbation and another for computing the gradient descent update (Liu et al., 2022). This has twice the computational overhead compared with SGD.

The effect of weight perturbation induced by device variability on model performance can also be considered a kind of generalization problem. We start with a perspective that the flat loss landscape will have robustness against variations in model parameters. A naive approach to solve device variability issues is applying techniques related to generalization. Unfortunately, as shown in Figure 1, these techniques did not lead to a significant improvement in weight variation. When there are variations in the weight, an accuracy drop of 36.58% for SGD, 34.17% for SWA, 27.44% for SAM, and 14.89% for AWP occurs compared with a case without variation. The accuracy drop caused by device variability is one of the major issues that make hardware implementation a challenge.

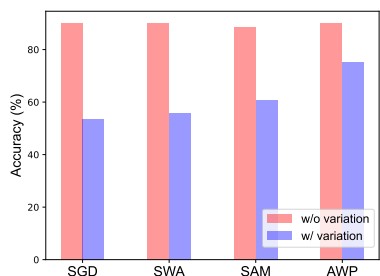

Figure 1: Accuracy of VGG-5 on CIFAR-10 using SGD, SWA, SAM, and AWP.

This motivates us to explore a new hardware-oriented neural network training method. Therefore, we proposed efficient and intuitive WVAT by considering stochastic weight characteristics incurred by device inherent variability in the training phase and in the weight update to flatten the loss landscape.

## 3 WEIGHT VARIATION-AWARE TRAINING (WVAT)

Our goal is to find parameters that are tolerant to perturbations of model parameters (e.g., weights) caused by the variability of synaptic hardware devices. We aim to minimize the loss, along with the difference in the loss with respect to perturbed weights. Thus, objective is as follows:

$$\min_{\boldsymbol{w}}[(L(\boldsymbol{w} + \boldsymbol{w_v}) - L(\boldsymbol{w})) + L(\boldsymbol{w})] \rightarrow \min_{\boldsymbol{w}} L(\boldsymbol{w} + \boldsymbol{w_v})$$

where where $\boldsymbol{w}$ and $\boldsymbol{w_v}$ denotes weight and weight variation, respectively. $L(\cdot)$ represents loss function. In order to explore the flat loss landscapes with respect to weight variation, the difference term $L(\boldsymbol{w} + \boldsymbol{w_v}) - L(\boldsymbol{w})$ should be minimized. When the weight trained in software (cloud) is transferred to a hardware device (edge device), the mapped weight is likely to be different from

the software weight due to device variability. This results in performance degradation. For these reasons, here we devise two types of variations that reproduce device variations during a training phase. One is a *hardware-simulated variation (HSV)* reflecting device variability, and the other is a *gradient-ascent variation (GAV)*.

### 3.1 HARDWARE-SIMULATED VARIATION (HSV)

Assume "analog" synaptic devices, as shown in Figure 2, the layer-wise weight range $A$ is set $[\mu - 3\sigma, \mu + 3\sigma]$ across the weight distribution, and we take a weight variation on $A$.

$$\Delta \boldsymbol{w} = \gamma \mathcal{N}(0, A\sigma_v{}^2)$$

where $\Delta \boldsymbol{w}$ is a *hardware-simulated variation (HSV)* to imitate intrinsic hardware device variability, which is a random sample of the same size as $\boldsymbol{w}$ from a Gaussian distribution with mean 0 and variation $A\sigma_v$. $\gamma$ is a range coefficient, scaling factor, to determine the variation size. In general, when mapping software weights to a synaptic hardware device, clipping method is widely used rather than the full range of software weights ($[w_{min}, w_{max}]$) due to the memory window limit of the synaptic device (Kwon et al., 2019; Joshi et al., 2020). Therefore, $A$ is the range of weights that can be expressed in synaptic hardware devices. HSV refers to how much variation occurs within the range that can be expressed in the device.

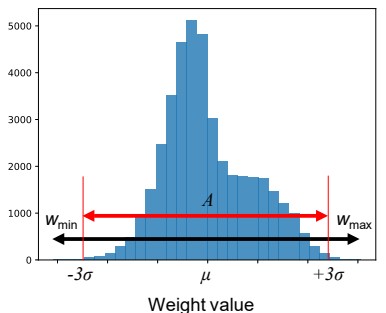

Figure 2: Weight distribution of a layer. A graph indicates the histogram of the software weight. Due to the device variability, the original software weight is perturbed. HSV is drawn from the weight range $A$ to mimic the variation.

Many studies have reported that fabricated synaptic devices have Gaussian distribution (Gong et al., 2018; Boybat et al., 2018; Yu et al., 2013); therefore, we use a Gaussian distribution when generating HSV. (However, noise may not follow normal distribution on real hardware devices. Further experiments on the distributions of real hardware devices and the noise distributions are in Section 4.5 and Appendix. A.5, respectively) Considering that the standard deviation of the fabricated devices is usually $\sim$5% (Joshi et al., 2020; Wan et al., 2019), we set that a 10% standard deviation was simulated during training for more stable results. For example, $\sigma_v$ of 10% means that the weight has changed by 10% of $A$. In order to minimize objective $L(\boldsymbol{w} + \Delta \boldsymbol{w})$ using SGD as an optimizer, the loss can be differentiated as follows:

$$\nabla_{\boldsymbol{w}} L(\boldsymbol{w} + \Delta \boldsymbol{w}) = \frac{d(\boldsymbol{w} + \Delta \boldsymbol{w})}{d\boldsymbol{w}} \frac{dL(\boldsymbol{w})}{d\boldsymbol{w}} \bigg|_{\boldsymbol{w} = \boldsymbol{w} + \Delta \boldsymbol{w}}$$
$$= \nabla_{\boldsymbol{w}} L(\boldsymbol{w})|_{\boldsymbol{w} = \boldsymbol{w} + \Delta \boldsymbol{w}}$$

$\nabla_{\boldsymbol{w}} L(\boldsymbol{w} + \Delta \boldsymbol{w})$ can be calculated as the differentiation at the value in which the weight variation occurs by the differentiation of the composite function.

### 3.2 GRADIENT-ASCENT VARIATION (GAV)

In addition to reproducing HSV during the training phase, the weight variation corresponding to the worst-case—making the greatest the difference term—should also be reflected to find flat loss landscapes. Recalling our objective, this is modified as a maximization problem.

$$\min_{\boldsymbol{w}} [L(\boldsymbol{w}) + \max_{\boldsymbol{v}(\boldsymbol{w})} (L(\boldsymbol{w} + \boldsymbol{w_v}) - L(\boldsymbol{w}))]$$
$$\rightarrow \min_{\boldsymbol{w}} \max_{\boldsymbol{w_v}} L(\boldsymbol{w} + \boldsymbol{w_v})$$

$L(\boldsymbol{w} + \boldsymbol{w_v})$ can be approximated by first-order Taylor expansion to find the weight variation $\boldsymbol{w_v}$ that maximizes the loss. For GAV, $\boldsymbol{w_v}$ is a function of $\boldsymbol{w}$, which can be expressed as follows:

$$L(\boldsymbol{w} + \boldsymbol{w_v}) = L(\boldsymbol{w} + \alpha \boldsymbol{v}(\boldsymbol{w}))$$
$$\approx L(\boldsymbol{w}) + \alpha \boldsymbol{v}(\boldsymbol{w})^T \nabla_{\boldsymbol{w}} L(\boldsymbol{w})$$

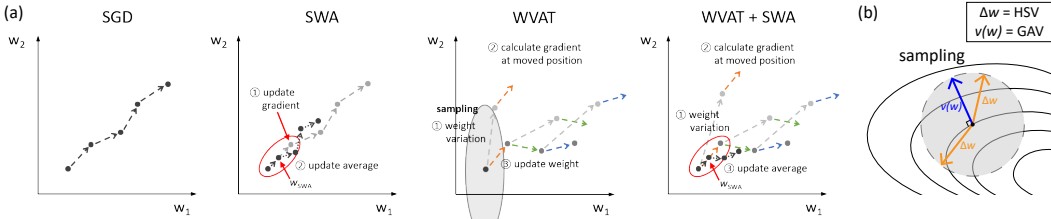

Figure 3: **(a)** Schematic of the weight update of the each method. **(b)** Schematice of the HSV and GAV. HSV reproduces in a randomized direction based on device variability, while GAV is the steepest ascent direction, which is the direction of weight perturbation in the worst case.

where $\alpha$ is a step-length parameter. $\boldsymbol{v}(\boldsymbol{w})^T \nabla_{\boldsymbol{w}} L(\boldsymbol{w})$ is the rate of change in $L$ along the direction $\boldsymbol{v}(\boldsymbol{w})$ at $\boldsymbol{w}$. Therefore, the most rapidly increasing direction is the solution to the problem.

$$\max_{\boldsymbol{v}(\boldsymbol{w})} \boldsymbol{v}(\boldsymbol{w})^T \nabla_{\boldsymbol{w}} L(\boldsymbol{w}) \quad \text{subject to } ||\boldsymbol{v}(\boldsymbol{w})|| = ||\Delta \boldsymbol{w}||$$

$$\boldsymbol{v}(\boldsymbol{w}) = \frac{\nabla L_{\boldsymbol{w}}(\boldsymbol{w})}{||\nabla L_{\boldsymbol{w}}(\boldsymbol{w})||} ||\Delta \boldsymbol{w}||$$

where the magnitude of the weight variation is set to be the same as that of HSV. $\nabla L_{\boldsymbol{w}}(\boldsymbol{w})$ is the steepest ascent direction for a line search method (Nocedal & Wright, 1999). For this reason, we name $\boldsymbol{v}(\boldsymbol{w})$ a *gradient-ascent variation (GAV)*. This is a similar approach to AWP and SAM.

### 3.3 WEIGHT UPDATE

Weight variation is simulated during training and updates the weight via SGD. The training process is as follows:

$$\text{Weight variation: } \boldsymbol{w} \leftarrow \boldsymbol{w} + \boldsymbol{w_v}$$
$$\text{Weight update: } \boldsymbol{w} \leftarrow \boldsymbol{w} - \eta \nabla L_{\boldsymbol{w}}(\boldsymbol{w} + \boldsymbol{w_v})$$
$$\text{Weight reverse: } \boldsymbol{w} \leftarrow \boldsymbol{w} - \boldsymbol{w_v}$$

where $\eta$ is a learning rate. In this case, $\boldsymbol{w_v}$ can be one of the two proposed HSV and GAV. Depending on the probability $p$, it determines whether or not to add $\boldsymbol{w_v}$ for each batch. If no variance is added, the weight update is equivalent to SGD, and no weight reverse is performed. We experimentally find that if $\boldsymbol{w_v}$ is larger than $\eta \nabla L$, it tends to diverge. $\gamma$ and $\alpha$ can be used to adjust the size of $\boldsymbol{w_v}$.

$$x \sim U(0, 1)$$
$$\boldsymbol{w_v} = \Delta \boldsymbol{w} \quad \text{if } x < w_{th}$$
$$\boldsymbol{w_v} = \boldsymbol{v}(\boldsymbol{w}) \quad \text{if } x \geq w_{th}$$

$w_{th}$ is a threshold of what kind of weight variation it will reproduce during training. $x$ is a randomly sampled value from a uniform distribution for each batch. For each batch, $x$ determines what kind of variation will be generated. Figure 3 schematically illustrates the weight update according to each method. SWA averages multiple points along the trajectory of SGD, leading to better generalization than SGD. By applying SWA to WVAT, wider loss landscapes can be explored.

## 4 EXPERIMENTS

In this section, we conduct experiments to evaluate the proposed WVAT on both artificial neural network (ANN) and spiking neural network (SNN) with different architectures, including fully connected (FC), convolutional neural network (CNN), VGG, ResNet, MobileNet, and EfficientNet on benchmark datasets (MNIST, N-MNIST, CIFAR-10, CIFAR-100, and ImageNet). These benchmarks and models have been widely used in hardware implementations (Kim et al., 2018; Long et al., 2019; Zhu et al., 2020; Joshi et al., 2020; Joksas et al., 2022; Huang et al., 2022; Jung et al., 2022). Edge devices are mainly implemented using small models; moreover, small models are vulnerable to performance degradation due to device variability. Hence, we focus on experiments on small models, including ablation studies and comparisons with SGD and SWA. Although SWA yields better performance via longer training, all method is trained for the same epochs for a fair comparison. We report the mean and standard deviation of test accuracy over 5 runs.

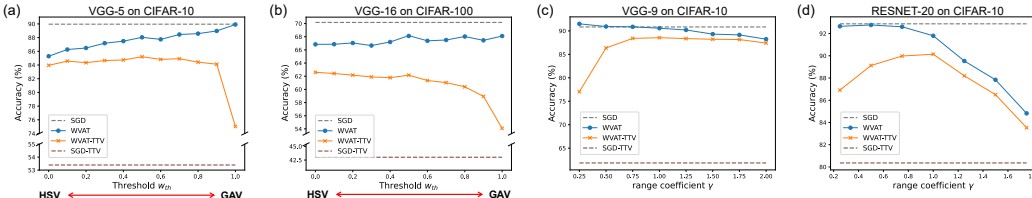

Figure 4: Accuracy of SGD and WVAT. (**a**), (**b**) Effect of HSV and GAV, and (**c**), (**d**) range coefficient $\gamma$ on model performance.

Figure 4 (a) and (b) shows the effect of two hyperparmeters on model performance for VGG-5 on CIFAR-10 and VGG-16 on CIFAR-100, respectively. We test WVAT for hyperparameter $w_{th} \in \{0, 0.1, 0.2, 0.3, 0.4, 0.5, 0.6, 0.7, 0.8, 0.9, 1\}$. If $w_{th} = 0$, only HSV is reflected, and if $w_{th} = 1$, only GAV is generated during training. For a baseline, we test SGD and SGD Test Time Variation (SGD-TTV), which means when the weight of the trained model perturbs during a test phase. As shown in Figure 4 (a), SGD achieved 90.09% accuracy, while SGD-TTV 53.51% when $\sigma_v$ of 10% (36.58% performance drop by the variation). In the case of $w_{th} = 1$, only GAV is reproduced during the training phase. Although it achieved the highest accuracy, it did not effectively prevent the degradation in accuracy when there was a variation in the weight. On the other hand, in the case of $w_{th} = 0$, WVAT minimized the degradation in accuracy during TTV, but the model performance was poor than SGD. Therefore, we set $w_{th} = 0.5$ as the default value, which achieves nearly the similar performance as SGD and is robust to weight perturbation induced by device variability.

Since HSV requires one gradient computation, like SGD, and GAV requires two sequential gradient computations, our proposed WVAT, which is a mixture of HSV and GAV, has less computational overhead compared to AWP and SAM. Training time using NVIDIA GPU Titan XP to train VGG-5 on CIFAR-10 is as follows. SGD took about 98.68 minutes, AWP ($w_{th} = 1$, only GAV) 147.11 minutes, and WVAT ($w_{th} = 0.5$) 114.42 minutes. Through this experiment, we experimentally confirmed that incorporating both HSV and GAV is a key factor in model performance and robustness against stochastic weight characteristics.

Range coefficient $\gamma$ is a hyperparameter that determines the variation size simulated in training. $\gamma = 1$ means that the variation size during TTV is equally reflected during training. As shown in Figure 4 (c) and (d), HSV with the same variation size as the test minimizes the accuracy drop. We set $\gamma = 1$ as the default value. Depending on the characteristics of each device, engineers can choose $w_{th}$ and $\gamma$ by considering the trade-offs for accuracy whether variations occur or not.

In terms of energy efficiency, it is ideal for a single device to represent a single synaptic value (Multi-bit per cell). We assume analog synaptic devices, but still, many devices are represented by multi-level states. Hence, we analyze the effect of quantization in Appendix A.3. We confirm that WVAT is comprehensively effective for implementing multi-bit cells with weight perturbations.

### 4.1 MNIST, N-MNIST

For MNIST and N-MNIST (Neuromorphic MNIST, Orchard et al. (2015)), we compared the accuracy of ANNs and SNNs in Appendix A.2. WVAT minimizes accuracy degradation from weight perturbations and achieves comparable performance to SGD and SWA when there are no weight perturbations. In SNNs, spatial and temporal integration exist, and the information is encoded by spike trains rather than values in ANNs. This is why SNNs are more vulnerable to weight variability. We find that WVAT works well with SNN training methods. In addition, WVAT does not require any changes to the training method itself, as it only needs to reproduce the weight perturbations during training. Therefore, there is no design or trick required for SNNs. This is one of the advantages of WVAT: it can be applied to a wide variety of cases without any customized tricks.

### 4.2 CIFAR-10

We test WVAT on CIFAR-10 in Table 1. For comparison experiments, we use VGG-5, VGG-16, and ResNet-110 models. While there are studies on how to train SNNs, converting trained ANNs to SNNs is still widely used due to high performance and limited resources. We adopt a hybrid conversion method (Rathi et al., 2020) for SNN. VGG-5 and VGG-16 are trained for 200 epochs using SGD with momentum 0.9, weight decay 0.0005, and an initial learning rate $lr$ of 0.01. We

Table 1: Accuracy on CIFAR-10 for different model and method.

| MODEL | METHOD | ACC (%) | ACC (%) $\sigma_v = 5\%$ | ACC (%) $\sigma_v = 10\%$ |
|---|---|---|---|---|
| VGG-5 (ANN) | SGD | $89.96 \pm 0.16$ | $79.08 \pm 0.95$ | $53.39 \pm 3.92$ |
| | SWA | $\mathbf{89.97} \pm 0.22$ | $80.62 \pm 1.51$ | $57.14 \pm 2.42$ |
| | WVAT | $87.76 \pm 0.24$ | $\mathbf{87.05} \pm 0.25$ | $\mathbf{85.21} \pm 0.19$ |
| VGG-16 (SNN CONVERSION) | SGD | $92.68 \pm 0.11$ | $90.49 \pm 0.08$ | $79.71 \pm 1.27$ |
| | SWA | $\mathbf{92.77} \pm 0.08$ | $91.05 \pm 0.16$ | $84.10 \pm 0.75$ |
| | WVAT | $91.79 \pm 0.14$ | $\mathbf{91.20} \pm 0.12$ | $\mathbf{89.25} \pm 0.10$ |
| RESNET-110 (ANN) | SGD | $95.06 \pm 0.17$ | $93.58 \pm 0.12$ | $84.68 \pm 0.71$ |
| | SWA | $\mathbf{95.47} \pm 0.06$ | $\mathbf{94.26} \pm 0.09$ | $87.26 \pm 0.95$ |
| | WVAT | $94.45 \pm 0.16$ | $93.64 \pm 0.13$ | $\mathbf{90.47} \pm 0.12$ |

Table 2: Accuracy on CIFAR-100 for different model and method.

| MODEL | METHOD | ACC (%) | ACC (%) $\sigma_v = 5\%$ | ACC (%) $\sigma_v = 10\%$ |
|---|---|---|---|---|
| VGG-16 (ANN) | SGD | $70.17 \pm 0.19$ | $63.88 \pm 0.41$ | $43.02 \pm 0.76$ |
| | SWA | $\mathbf{70.46} \pm 0.32$ | $66.33 \pm 0.25$ | $51.87 \pm 1.25$ |
| | WVAT | $68.13 \pm 0.12$ | $\mathbf{66.51} \pm 0.14$ | $\mathbf{62.16} \pm 0.11$ |
| WRN-28-10 (ANN) | SGD | $80.62 \pm 0.21$ | $78.83 \pm 1.05$ | $69.37 \pm 2.90$ |
| | SAM | $\mathbf{82.31} \pm 0.20$ | $79.25 \pm 0.38$ | $71.88 \pm 2.22$ |
| | SWA | $82.12 \pm 0.18$ | $80.00 \pm 0.44$ | $74.07 \pm 1.97$ |
| | WVAT | $81.26 \pm 0.34$ | $\mathbf{80.23} \pm 0.19$ | $\mathbf{76.53} \pm 0.79$ |
| RESNET-110 (ANN) | SGD | $76.72 \pm 0.55$ | $69.22 \pm 0.42$ | $32.68 \pm 0.75$ |
| | SWA | $\mathbf{78.52} \pm 0.19$ | $74.33 \pm 0.14$ | $52.53 \pm 0.32$ |
| | WVAT | $78.13 \pm 0.94$ | $\mathbf{74.59} \pm 0.87$ | $\mathbf{54.25} \pm 0.82$ |

use Preactivation ResNet-110 in Garipov et al. (2018). ResNet-110 are trained for 150 epochs using SGD with momentum 0.9, weight decay 0.0003, and an initial $lr$ of 0.1. When applying SWA, we first run SGD optimizer with a decaying $lr$ schedule for 75% of the training budget, and then apply SWA with a fixed $lr$ of 0.005 for all models except for 0.01 for ResNet-110. Hyperparameters for WVAT are set as $p = 1$, $\gamma = 1$, $w_{th} = 0.5$, and $\alpha = 0.01$ except for $p = 0.5$ for ResNet-110.

We experimentally confirm that WVAT minimizes the decrease in accuracy when there is variation induced by device variability in most models. Compared to MNIST, the performance degradation due to weight perturbation is more pronounced. In the case of VGG-5, when $\sigma_v = 10\%$, WVAT yields 85.21% accuracy, while 53.39% for SGD and 57.14% for SWA. The accuracy drop is 36.57% for SGD, 32.83% for SWA, and 2.55% for WVAT. We demonstrated that the advantage of WVAT in small models is more significant.

### 4.3 CIFAR-100

We compare the accuracy of each method on CIFAR-100, and the results are summarized in Table 2. For comparison experiments, we use VGG-16, WideResNet(WRN)-28-10, and ResNet-110 models. VGG-16 is the same as the experimental setting of CIFAR-10. WRN-28-10 is trained for 200 epochs using SGD with momentum 0.9, weight decay 0.0005, and an initial $lr$ of 0.1. ResNet-110 is trained for 150 epochs using SGD with momentum 0.9, weight decay 0.0003, and an initial $lr$ of 0.05. For SWA settings, same as CIFAR-10. Hyperparameters for WVAT are set as $p = 0.5$, $\gamma = 1$, $w_{th} = 0.5$ for all models, and $\alpha = 0.01$ for VGG-16, $\alpha = 0.05$ for WRN-28-10, and $\alpha = 0.1$ for ResNet-110.

A comparison experiment is conducted with SAM, a method similar to GAV, using the WRN-28-10. When $\sigma_v = 10\%$, WVAT yields 76.53% accuracy, and SAM does 71.88%. It is confirmed that WVAT is more robust to weight perturbations than SAM. As mentioned in Section 4.2, WVAT outperforms other methods not only in the small model but also in ResNet-110. It is important to note that the standard deviation of accuracy for WVAT is also the smallest for each network architecture, which means that the proposed WVAT is tolerant to weight variation. For example, when $\sigma_v = 10\%$ on VGG-16, WVAT achieves $(62.16 \pm \mathbf{0.11})$ %, $(43.02 \pm \mathbf{0.76})$ % for SGD, and $(51.87 \pm \mathbf{1.25})$ % for SWA.

### 4.4 IMAGENET

We test each method on ImageNet (Russakovsky et al., 2015). We use ResNet-34, MobileNetV2 (Sandler et al., 2019), and EfficientNet (Tan & Le, 2020) for comparison experiments, which are pre-trained models provided by PyTorch. For WVAT training, ResNet-34 is retrained for 10 epochs with similar settings as (Joshi et al., 2020). The top-1 accuracy of ResNet-34 is shown in Figure 5. HSV mimics the variability that occurs in hardware. Joshi et al. (2020) injects Gaussian noise during training, which is the same way as HSV. The accuracy of the pre-trained ResNet-34 model using SGD is 73.31%. When $\sigma_v =$

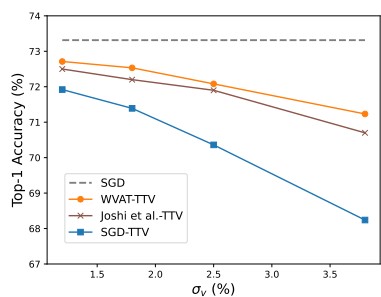

Figure 5: ResNet-34 on ImageNet.

3.8%, WVAT yields 71.23% accuracy, while 68.24% for SGD and 70.7% for Joshi et al. (2020). We find the accuracy on ImageNet to be more vulnerable to weight variability. Through comparison with SAM in CIFAR-100 and comparison with (Joshi et al., 2020) in ImageNet, it is again confirmed that incorporating both GAV and HSV in training is significant in the robustness of the model with regard to weight perturbations. This is in line with the experiments in Figure 4 (a). Results for MobileNetV2 and EfficientNet and detailed experimental settings are in Appendix A.4.

### 4.5 REAL HARDWARE DEVICE

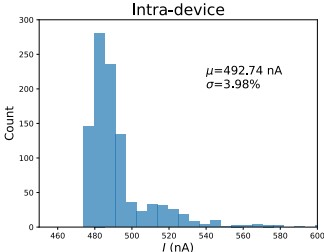 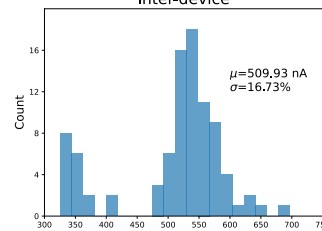 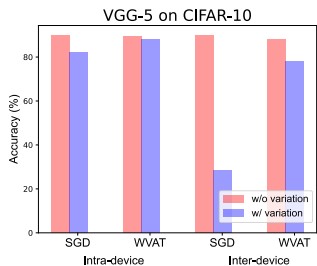

Figure 6: Tunneling current distribution of ferroelectric tunnel junctions (FTJ) at low resistance state (LRS). (**left**) Intra-device distribution (**middle**) Inter-device distribution. (**right**) Accuracy of VGG-5 on CIFAR-10 on FTJ decives.

Since the noise in a real device can have other distributions besides the normal distribution, we conducted experiments using the distributions of real hardware devices. We investigate the impact of device variability on neural network performance using measurements of ferroelectric tunnel junctions (FTJ), the neuromorphic synaptic devices fabricated by our group. Figure 6 shows the measurements of the fabricated devices. Where intra-device distribution is 1000 repeated measurements of low resistance state (LRS) tunneling current in a single FTJ device, and inter-device distribution is the same measurement on 100 different devices. HSV is reproduced by following the measured FTJ device distributions during training, which is the same during test time.

The experimental results of VGG-5 on CIFAR-10 are shown on the right side of Figure 6. For the intra-device, WVAT achieves an accuracy of 88.14%, while 82.06% for SGD with variation. For the inter-device, WVAT is 78.01%, and SGD is 28.43%, with WVAT outperforming by a large margin. The intra-device distribution of the FTJ device has a variance of 3.83% and a right-skewed normal distribution. However, the inter-device distribution has a multimodal distribution, which makes SGD more vulnerable to performance degradation due to weight perturbation by inherent device variability. However, WVAT shows it is robust to weight variation even with a multimodal distribution. WVAT is also effective for other noise distributions (see Appendix A.5).

## 5 DISCUSSION

### 5.1 WEIGHT LOSS LANDSCAPES

In this section, an experiment is first conducted to investigate the effect of the proposed WVAT on the geometry of *weight loss landscapes*. We visualize the loss landscapes by plotting the change in loss as the weight moves along the direction of the weight variation $\Delta \boldsymbol{w}$.

$$L(\boldsymbol{w} + \Delta \boldsymbol{w}) = L(\boldsymbol{w} + \gamma \mathcal{N}(0,\ A{\sigma_v}^2))$$

where $\sigma_v$ varies in the range of $[0, 15]\%$, and moving direction is adjusted with $\gamma = -1$ and $\gamma = 1$. We visualize it as $\sigma_v$ drawn from 10 different samplings.

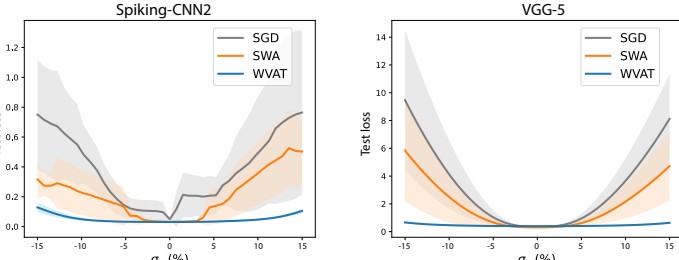

Figure 7: Loss landscapes with respect to model weights. (**left**) Spiking-CNN2 on MNIST, and (**right**) VGG-5 on CIFAR-10.

Figure 7 shows the test loss as a function of the magnitude of the weight variation. Weight loss landscapes become broader and flatter in the order of SGD, SWA, and WVAT, and are uniform for multiple trials. We are taken together with the experimental results in Section 4; these experiments show the connection between flat weight loss landscapes and robustness to weight perturbations.

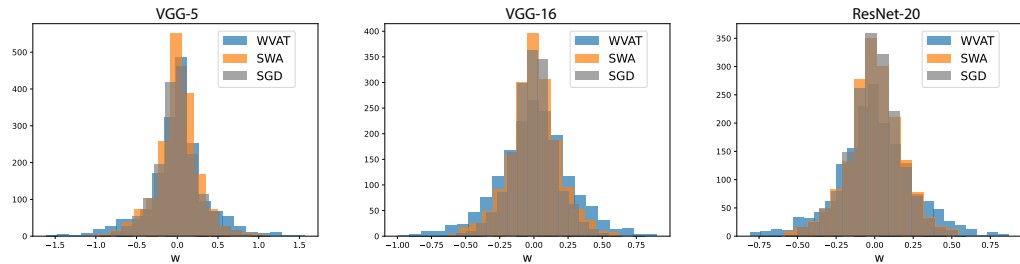

Figure 8: Weight distribution of the first convolutional layer on CIFAR-10.

Secondly, we carefully investigate the effect of the proposed WVAT on the weight distribution. As shown in Figure 8, we verify that WVAT produces larger weights than SGD and SWA (The weight distribution over the entire network is in Appendix A.6). Li et al. (2018) argued that small weights are more sensitive to weight perturbations and make a sharper loss landscape. This claim is consistent with our experimental results. Thus, based on these observations, we demonstrate that WVAT produces larger weights, which makes it less sensitive to weight perturbations and leads to a flatness of the weight loss landscapes.

## 6 CONCLUSION

This paper proposes a weight variation-aware training method that is robust to weight perturbations incurred by device variability. For the first time to our knowledge, we investigate and analyze the impacts of weight variations on various benchmark datasets and network architectures. The proposed WVAT effectively minimizes performance degradation by more than 1/10 compared to SGD when there is a weight variation. We propose HSV and GAV to mimic weight variations during the training and present a weight update method that considers the stochastic weight characteristics. We experimentally confirm that WVAT is tolerant to weight perturbations by finding the flat loss landscapes with respect to weight. This method is a hardware-oriented training method at the algorithm level rather than a custom solution at the device level to reduce the performance degradation for the stochastic weight characteristic caused by the inherent variability of the device. Therefore, when the weights trained by WVAT are transferred to hardware, accuracy drop due to device variability can be prevented. It is especially effective in small models.

For limitations and future works, WVAT computes the gradient-ascent variation, requiring two sequential gradient computations, like AWP and SAM, so there is room for computational improvement. Further research is needed on how to achieve similar performance to WVAT with less computational overhead. We think that studies like Liu et al. (2022) can inspire us to solve this problem.

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

# A APPENDIX

## A.1 COMPARISON WITH OTHER VARIATION-AWARE METHODS

The approach of Liu et al. (2015); Zhu et al. (2020) is the customized solutions for the given device, and Long et al. (2019) (1 bit per cell) is different from the analog synapse, which is the main target of the paper. Although these methods do not apply to the general case, it is meaningful to compare other variation-aware methods, so we conducted comparison experiments.

We conducted the comparison experiments on a 2-layer fully-connected network on MNIST, the same in Liu et al. (2015); Zhu et al. (2020), and AlexNet on CIFAR-10, the same in Long et al. (2019), respectively. Table 3 and Figure 9 shows comparison results. These results show that WVAT is more robust to weight variation than other variation-aware methods.

Table 3: Accuracy in a 2-layer fully-connected network on MNIST.

| METHOD | ACC (%) | ACC (%) $\sigma_v = 10\%$ |
|---|---|---|
| SGD | 97 | $32 \pm 3.3$ |
| LIU ET AL. (2015) | 92 | $38 \pm 3.2$ |
| ZHU ET AL. (2020) | 92 | $92 \pm 1.0$ |
| WVAT | 95 | $93 \pm 1.0$ |

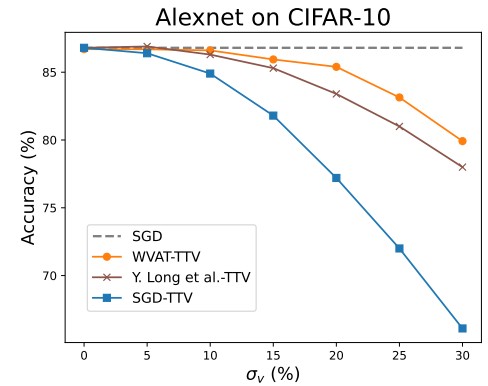

Figure 9: Accuracy of AlexNet on CIFAR-10.

## A.2 MNIST, N-MNIST

For MNIST and N-MNIST, we compare the accuracy of the FC network and spiking CNN trained by each method in Table 4. We experimented with SLAYER (Shrestha & Orchard, 2018) and TSSL-BP (Zhang & Li, 2020), a method for training SNNs. We train the FC network by (1) using an SGD optimizer to train an ANN on the MNIST and (2) using an Adam optimizer with SLAYER to train an SNN on N-MNIST. We also train spiking-CNNs by TSSL-BP for training SNN on MNIST. All models are trained for 100 epochs. The learning rate is 0.0005 for ANN. For SLAYER and TSSL-BP, we use Adam optimizer with a learning rate of 0.001 and 0.0005, respectively, and other settings are the same as in Shrestha & Orchard (2018); Zhang & Li (2020). Hyperparameters for WVAT are set as default value, $p = 1$, $\gamma = 1$, $w_{th} = 0.5$, and $\alpha = 0.1$, except for $\alpha = 0.05$ for SLAYER. When applying SWA, we use SWA training with a fixed learning rate schedule from scratch.

For each network architecture and method, WVAT achieves nearly similar performance as SGD and SWA and minimizes accuracy degradation when weight variation occurs. Hardware devices inevitably have inherent variability. Therefore, in most cases, the model performance in software cannot be directly reproduced in hardware implementation. From the hardware implementation point of view, it is preferable to refer to the accuracy when there is weight variation rather than the accuracy when there is no variation. In the case of spiking-CNN1, when $\sigma_v = 10\%$, WVAT

substantially reduces the performance drop compared with TSSL-BP (3.29% performance drop for WVAT, 16.97% for TSSL-BP).

Table 4: Accuracy on MNIST and N-MNIST for different model and method. 15C5 means convolution layer with 15 of the $5 \times 5$ filters, and P2 pooling layer with $2 \times 2$ filters.

| MODEL | METHOD | ACC (%) | ACC (%) $\sigma_v = 5\%$ | ACC (%) $\sigma_v = 10\%$ |
|---|---|---|---|---|
| FC 2-LAYER[1] (ANN, MNIST) | SGD | $98.13 \pm 0.09$ | $97.86 \pm 0.14$ | $96.54 \pm 0.23$ |
| | SWA | $\mathbf{98.56} \pm 0.12$ | $98.31 \pm 0.71$ | $97.20 \pm 1.54$ |
| | WVAT | $98.47 \pm 0.05$ | $\mathbf{98.39} \pm 0.04$ | $\mathbf{97.98} \pm 0.19$ |
| FC 2-LAYER[1] (SNN, N-MNIST) | SLAYER | $\mathbf{98.64} \pm 0.04$ | $72.01 \pm 0.13$ | $38.76 \pm 15.43$ |
| | SWA | $98.61 \pm 0.06$ | $78.97 \pm 9.08$ | $39.83 \pm 13.50$ |
| | WVAT | $98.02 \pm 0.05$ | $\mathbf{97.48} \pm 0.13$ | $\mathbf{96.29} \pm 0.63$ |
| SPIKING-CNN1[2] (SNN, MNIST) | TSSL-BP | $\mathbf{99.37} \pm 0.02$ | $98.08 \pm 0.15$ | $82.40 \pm 8.03$ |
| | SWA | $99.29 \pm 0.11$ | $98.62 \pm 0.60$ | $90.24 \pm 2.34$ |
| | WVAT | $99.31 \pm 0.07$ | $\mathbf{99.07} \pm 0.07$ | $\mathbf{96.02} \pm 0.82$ |
| SPIKING-CNN2[3] (SNN, MNIST) | TSSL-BP | $99.38 \pm 0.05$ | $98.81 \pm 0.06$ | $86.17 \pm 8.55$ |
| | SWA | $\mathbf{99.44} \pm 0.03$ | $98.23 \pm 0.08$ | $89.73 \pm 2.69$ |
| | WVAT | $99.39 \pm 0.03$ | $\mathbf{99.24} \pm 0.08$ | $\mathbf{95.57} \pm 1.87$ |

[1] 784-400-200-10.
[2] 16C5-P2-CONCAT(32C3, 8C1)-8C1-288.
[3] 15C5-P2-40C5-P2-300.

## A.3 ROBUSTNESS TO QUANTIZATION

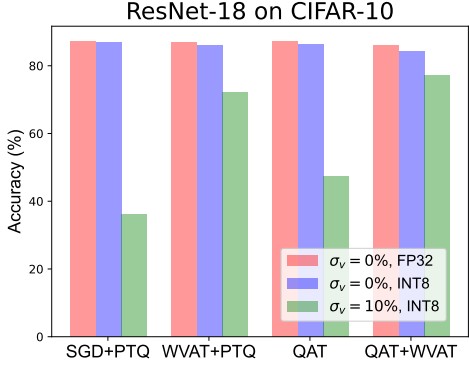

Figure 10: Accuracy of ResNet-18 on CIFAR-10.

We assume analog synapses in the main paper, but many synapses still represent multi-bit weights. Although WVAT is not for a quantization technique, the fact that WVAT is robust to weight perturbation implies that it is also effective against quantization, another hardware implementation issue. Therefore, we conduct experiments to verify the validity of WVAT when quantization is applied using ResNet-18 on CIFAR-10.

Figure 10 shows the comparison results of two widely-used quantization methods, Post-Training Quantization (PTQ) and Quantization-Aware Training (QAT). The case of training with SGD and applying PTQ is called SGD+PTG, and the case of training with WVAT and applying PTQ is called WVAT+PTQ. When $\sigma_v = 10\%$, SGD+PTQ achieves 87.06% accuracy and 50.93% accuracy drop on int8, while WVAT+PTQ has 85.97% accuracy and 13.86% accuracy drop on int8, respectively. QAT achieves 86.49% accuracy and 39.06% accuracy drop on int8, while QAT+WVAT has 84.41% accuracy and 7.06% accuracy drop on int8, respectively. We experimentally confirm that WVAT

is more robust to weight variation in quantization. WVAT is effective at both analog synapses and multi-level state synapses, making the method applicable to a wide range of neuromorphic implementation scenarios.

## A.4 IMAGENET

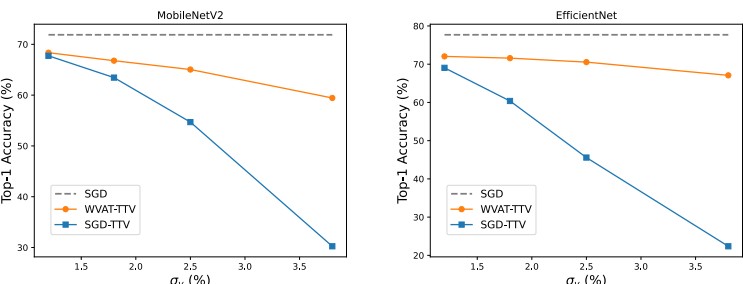

Figure 11: Training results on (**left**) MobileNetV2 and (**right**) EfficientNet on ImageNet.

We test each method on ImageNet (Russakovsky et al., 2015). For WVAT training, we train ResNet-34 model using similar settings as Joshi et al. (2020); ResNet-34 is fine-tuned for 10 epochs using SGD with momentum 0.9, weight decay 0.0001, and a constant $lr$ of 0.00005. Hyperparameters for WVAT are set as $p = 1.0$, $\gamma = 0.833$, $w_{th} = 0.5$, and $\alpha = 0.005$ for ResNet-34. For comparison with (Joshi et al., 2020), the weight range is adjusted to be the same as (Joshi et al., 2020), and $\gamma = 0.833$.

We use pre-trained MobileNetV2 and EfficientNet provided by PyTorch model zoo. MobileNetV2 is also fine-tuned for 10 epochs using SGD with momentum 0.9 and weight decay 0.0001. It used StepLR, a learning rate scheduler with an initial learning rate of 0.0001 and a multiplicative factor of a learning rate of 0.95 for every epoch. Hyperparameters for WVAT are the same as those of ResNet-34, except for $\gamma = 1.0$ for MobileNetV2. EfficientNet is fine-tuned for 5 epochs using RMSprop with momentum 0.9 and weight decay $10^{-5}$. It used ExponentialLR, a learning rate scheduler with an initial learning rate of 0.0001 and a multiplicative factor of a learning rate of 0.965. Hyperparameters for WVAT are the same as those of MobileNetV2.

The accuracies of the pre-trained MobileNetV2 and EfficientNet models are 71.88% and 77.69%, respectively. In case of MobileNetV2, when $\sigma_v = 3.8\%$, WVAT yields 59.44% accuracy, while 30.24% for SGD. In case of EfficientNet, when $\sigma_v = 3.8\%$, WVAT yields 67.08% accuracy, while 22.37% for SGD. We confirm that similar to CIFAR-10 and CIFAR-100, the proposed method also prevents accuracy degradation for weight variations on ImageNet.

## A.5 EFFECT ON NOISE DISTRIBUTION

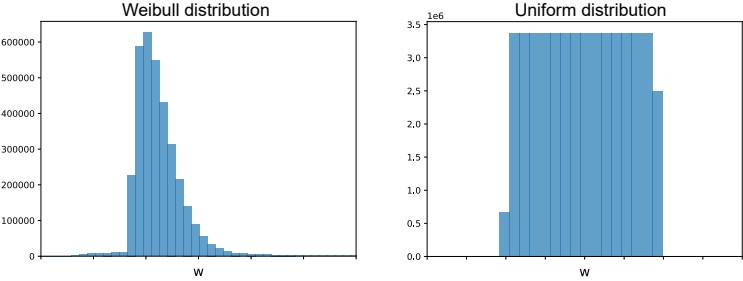

Figure 12: Noise Distribution. (**left**) Weibull distribution (**right**) Uniform distribution.

As mentioned in Section 3.1, we experimented by inserting Gaussian noise, which was effective in real devices (Joshi et al., 2020). However, noise in real hardware devices may not follow Gaussian

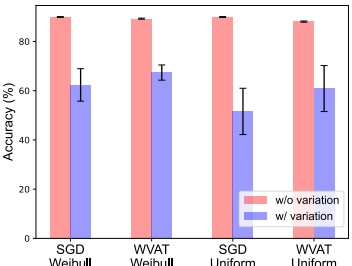 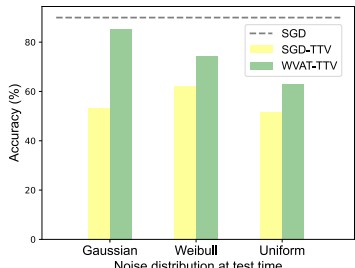

Figure 13: (**left**) Accuracy of ResNet-18 on CIFAR-10 when the same noise distribution occurs in training and test. (**right**) Results of generating Gaussian noise distribution during training and different noise distributions during the test.

distribution. For this reason, an experiment is conducted to verify if the proposed method is effective even when the noises follow a distribution other than the Gaussian distribution. We assume that the noise follows the Weibull distribution, which is a right-skewed distribution, and a uniform distribution, which can be considered the worst-case. Figure 12 shows each noise distribution.

The left side of Figure 13 shows the results when the same noise is used for training and test time (inference), while the right side shows the results when Gaussian noise is used for training and a different type of noise is used for test. As shown in Figure 13 (left), in the case of Weibull distribution, an accuracy drop of 27.60% for SGD and 21.90% for WVAT occurs compared with a case without variation. When noise follows uniform distribution, accuracy drops are 38.38% for SGD and 27.22% for WVAT. We confirm that WVAT is effective against weight variations even when the noise follows a distribution other than the ideal Gaussian distribution.

Moreover, Figure 13 (right) shows that it is still robust to weight variations even when trained with Gaussian distribution and then follows a distribution different from the Gaussian distribution at test time. We experimentally confirm that WVAT is effective not only in Gaussian distribution but also in other types of distribution. Therefore, if real hardware devices do not follow Gaussian distribution, reproducing the distribution of real hardware devices when generating the HSV can still obtain a model that is robust to device variability. From this, we demonstrate that incorporating the effects of weight perturbations at inference during the training is a more critical factor than the shape of the noise distribution. Therefore, if the distribution of the real device is known, it is best to reproduce that distribution during training; if not, users can still get a model that is robust to weight variations by training under the assumption that the noise follows a Gaussian distribution.

### A.6 WEIGHT DISTRIBUTION

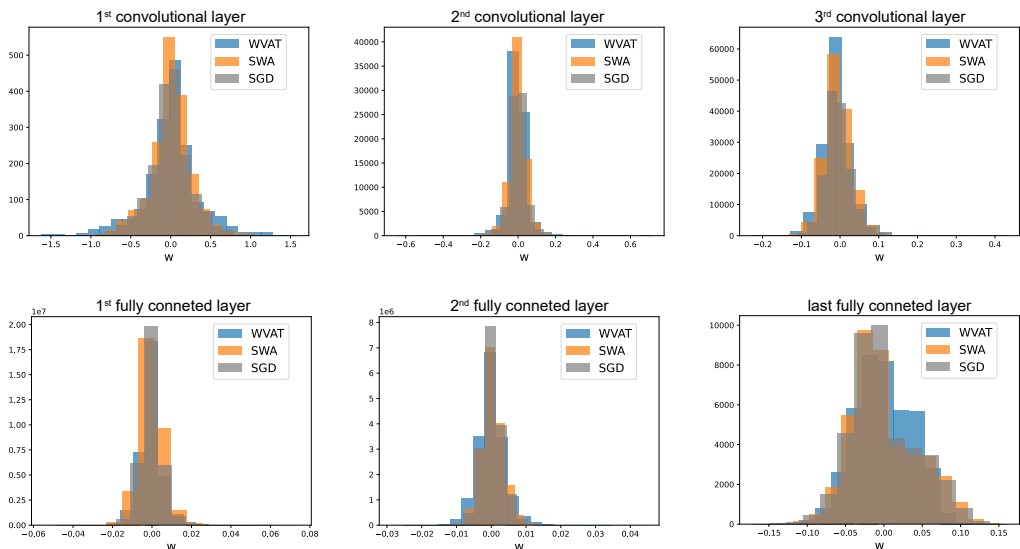

Figure 14: Weight distribution of the entire network layer of VGG-5 on CIFAR-10.

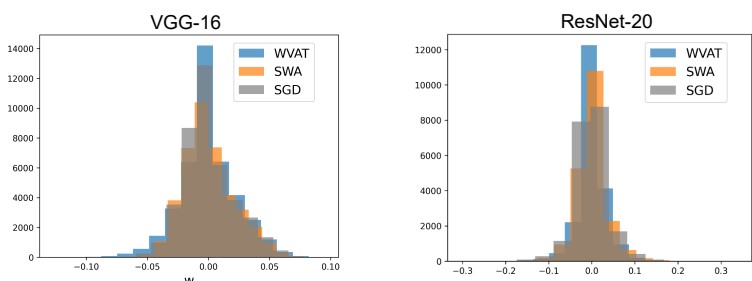

Figure 15: Weight distribution of the last FC layer of VGG-16 and ResNet-20 on CIFAR-10.

