# OpenReview forum: "A Weight Variation-Aware Training Method for Hardware Neuromorphic Chips"
_ICLR.cc/2024/Conference — ICLR 2024 Conference Withdrawn Submission_

### Official Review · Reviewer_tUb1 · 2023-10-29

**Soundness:** 3 good
**Presentation:** 3 good
**Contribution:** 2 fair
**Rating:** 3
**Confidence:** 5

**Summary:**

This paper proposes to combine noise-aware training and sharpness-aware minimization to improve the weight-noise robustness of DNNs for both ANN and SNN. It shows higher robustness to device variation then SGD and SWA.

**Strengths:**

Strength:

1.	It combines HSV and GAV during training to improve robustness and evaluated on different datasets/models.

**Weaknesses:**

Weaknesses:

1.	The novelty of the proposed method is very limited. Noise-aware training with noise injection, using SAM for landscape smoothing for analog neural networks with random noises, and device variations are standard.

2.	The noise-aware training performance heavily relies on accurate noise variation modeling, which is why Fig 4 shows the performance is the best when gamma is 1, which is also a well-known conclusion in analog NNs.

3.	The proposed method sacrifices accuracy when noise is small, e.g., Table 1 and 2. The fundamental reason is that the noise injection impacts the convergence. A more advanced method is to use noise source cooling to avoid noise-induced convergence issues, which can make it less sensitive to the actual noise intensity.

4.	The paper claims to be the first one to investigate device variation in analog neural networks, SNNs. As far as I know, there are many robustness-driven optimization, noise-aware training, on-chip training, and physical training methods in the literature to solve this problem, while none of those prior methods are reviewed and compared.

**Questions:**

same as weaknesses.

---

### Official Review · Reviewer_Z5Eu · 2023-10-30

**Soundness:** 2 fair
**Presentation:** 3 good
**Contribution:** 2 fair
**Rating:** 3
**Confidence:** 5

**Summary:**

This paper aims to address the accuracy degradation issues in neuromorphic chips posed by weight instability. The authors investigate the accuracy loss due to weight variation and propose to seek a flat weight loss landscape to preserve accuracy under weight variations. Two techniques are illustrated to inject noise during the normal training process for this target, including hardware-simulated sampled noise (WAV) and artificially calculated gradient-ascent variation (GAV).

Various experiments are conducted on different architectures and datasets on both ANN and SNN. They claim the proposed method can dramatically minimize the performance drop

**Strengths:**

- Extensive experiments evaluation, including both SNN and ANN.
- The easy-to-read technical part.

**Weaknesses:**

- Limited novelty on the technical part. The weight variation-aware training is not new in the neuromorphic area, where we typically inject hardware-simulated variation (your HSV) with the same process in 3.3. The GAV is a similar approach to AWP and SAM. The weight update scheme is not new, and the variation injection scheme is a simple mixture of GAV and HSV.
- Improper baselines. The authors mainly compare their method with vanilla SGD without considering the noises, making the claimed performance improvement intriguing. It is not a fair comparison. The prior noise-aware training technique should be used as the baseline in all experiments.

**Questions:**

Could you elaborate on the training details of variation-aware training in A.1. What’s the amount of noise you injected in noise-aware training?

---

### Official Review · Reviewer_WCbC · 2023-10-31

**Soundness:** 3 good
**Presentation:** 2 fair
**Contribution:** 2 fair
**Rating:** 5
**Confidence:** 5

**Summary:**

This paper proposes a new training method with the aim of making the parameters of the trained network model robust to random perturbations, with the perspective of writing the trained weights on noisy memory devices such as Ferroelectric Tunnel Junctions for neuromorphic computing applications. More specifically, the paper proposes two "weight variation-aware training" (WVAT) methods, "hardware-simulated variation" (HSV), and "gradient-ascent variation" (GAV). HSV samples the perturbation from a Gaussian distribution, while GAV is the "worst case" scenario when the perturbation is put in the direction of ascending the loss gradient. The weight update proposed consists in computing the loss gradient at the perturbed weights and applying to the unperturbed weights, with a stochastic threshold for choosing between HSV and GAV. The method is tested over a wide range of network architectures and datasets, where the test accuracy is reported for different amount of perturbations. Overall WVAT diminishes the peak performance without noise, but provides robustness to noise (reduces the accuracy degradation in the presence of noise) compared to noiseless SGD. The authors also use noise taken from the distribution of actual FTJ devices, and report similar findings. An analysis of the loss landscape is provided by computing the loss of the perturbed parameters, showing that WVAT finds flat minima of the loss landscape.

**Strengths:**

The strengths of the paper are:

**Simplicity**: The method proposed is simple and effective in providing noise robustness.

**Exhaustive**: The method is tested on a large array of tasks and architectures, and averaged over multiple seeds.

**Interdisciplinary**: The papers touches on concepts from pure ML (generalization methods) and neuromorphic computing (FTJ devices)

**Weaknesses:**

The weaknesses of the paper are:

**Novelty**: The main finding of the paper is that when noise is accounted for during training, the final model is more robust to noise, albeit less performant in the noiseless case. This is a rather known pattern in the neuromorphic literature, see e.g. [1] (section IV) or [2]. From this point of view, the finding that adding noise during training yields noise robustness is not very novel, but consistent with prior work.

**Ablation**: The proposed method is introduced in a principled way, which is nice, but what about removing the "Weight reverse" step? I wonder whether this even simpler approach would give similar result or not. This would shed light on whether the theory is both necessary and sufficient or just sufficient. Testing whether using Dropout during training gives any robustness would be also interesting.

Overall, the paper is ok, but I find the contribution is a bit weak for the general audience of ICLR. I think this paper would be well-suited for a more specific venue. For this reason, I will recommend weak reject for now, with the possibility to revise my judgement if my points are sufficiently addressed.

___
[1] Hirtzlin, Tifenn, et al. "Outstanding bit error tolerance of resistive ram-based binarized neural networks." 2019 IEEE International Conference on Artificial Intelligence Circuits and Systems (AICAS). IEEE, 2019.
[2] Rasch, Malte J., et al. "Hardware-aware training for large-scale and diverse deep learning inference workloads using in-memory computing-based accelerators." Nature Communications 14.1 (2023): 5282.

**Questions:**

- What is the robustness of the network when the "Weight reverse" step is removed? (duplicate from the above section).

- Does using dropout during training help robustness and to which extent? (duplicate from the above section).

- Do the network use batch normalization?

- Section 4.5:
    - Could the authors explain how the FTJ current distributions shown in the low resistance state (LRS) relate to the reading of the parameters in a neuromorphic chip? A section in the appendix detailing the use of the FTJ to encode the weights would be beneficial.
    - How is the current on the x-axis related to the value of a weight in the network?
    - Shouldn't the high resistance state (HRS) distributions also be used to encode the weight?
    - Shouldn't both the intra and inter device variability be taken into account when evaluating the robustness? Right now they seem to be used separately.

---

### Official Review · Reviewer_Q4cw · 2023-10-31

**Soundness:** 2 fair
**Presentation:** 2 fair
**Contribution:** 1 poor
**Rating:** 3
**Confidence:** 4

**Summary:**

The authors describe a method for hardware-aware re-training of DNNs
and SNNs (called WVAT), to be deployed on in-memory (noisy) analog AI hardware
inference accelerators and low-power neuromorphic hardware designed
for edge AI. After a general introduction of related methods, the WVAT
method is introduced by a min-max optimization (GAV), where the optimization
landscape is forced to be flat (to support robustness to weight
variations). In more detail, in two-gradient passes, first the weight
gradient is computed (on the "reference" weights) and the a second
weight gradient is computed in an (approximation of the ) worst case
direction, which turns out to be in the direction of the first
gradient. This is used in combination with random weight directions (HSV),
where one of the approaches is applied for a given mini-batch.

The authors show that the resulting trained method is more robust than
a DNN trained with SWA (stochastic weight averaging) or standard SGD on a
number of DNNs on image classification benchmarks when weight noise is
present.

They also show that the WVAT-trained DNNs are more robust to
hardware-simulated weight noise distributions (instead of generic
Gaussians).

**Strengths:**

* The general topic is interesting and important for successful deployment of DNNs on neuromorpiic chips.  The proposed WVAT method is an interesting idea based on the flatness of the loss landscape.

* Weight variation seen in an hardware experiment was tested.

* The authors show that robustness is also increased for SNNs, which
seems novel. However, it is not surprising that if a DNN is more noise
robust against weight variations, its corresponding converted SNN
should be more robust as well, so maybe this aspect is not interesting
enough in itself.

**Weaknesses:**

* While the study is reasonable well executed and written, the novelty
and improvements over state-of-the-art methods is very limited.  In
particular, the authors claim to "for the first time" analyze the
impacts of weight variations for various DNNs and benchmarks which is
not correct. In a recent study (Rasch et al. 2023 Nat Comm) -- which
the authors fail to cite and thus might be not aware of -- a much more
rigorous investigation is done over many DNNs and benchmarks,
including not only CNNs but also transformers and LSTMs. That study
also improves on the Joshi et al 2020 study in terms of achieved
robustness. So the study of Joshi et al, which the authors compare
against, cannot be considered "state-of-the-art" anymore. Moreover,
the proposed WVAT method does not even seem to convincingly improve
upon Joshi et al (see below) although using twice as long computation
(two gradient passes needed).

* Even against the Joshi et al 2020, which is in fact the
same HSV method that is part of the WVAT method (as the authors
acknowledge), the robustness improvement is very limited if any. In
fact, when varying from purely HSV (as in Joshi et al) to purely GAV,
then Figure 4ab shows that purely HSV is indeed the most robust
method, and not WVAT (see in partcilar 4b). This is not discussed and it
seems that the results of Figure 4 were misinterpreted: note that when
comparing the methods, the non-noisy accuracy is completely irrelevant,
since DNNs are to be deployed on noisy hardware. Thus, the methods
should be compared based on the "TTV" accuracy only. Here, in
particular in Figure 4b,  the 100% HSV method is actually always
better than adding GAV. This is what Joshi et al already used. Thus, I don't
see the additional benefit of the new method.

* The experimental section compares the WVAT method against SWA and
SGD, however, these are not methods designed to improve robustness
for noisy hardware. Instead, it should at least be compared
against 100% HSV, which is what e.g. Joshi et al. (or in a more
sophisticated manner with more realistic noise distributions and
hardware assumptions, Rasch et al. 2023) suggested. Figure 4b indicates
that 100% HSV would indeed be better or at least very similar to
WVAT (see above). This comparison is missing.

* It is also not clear why the additional SAM and AWP  methods
are introduced in the beginning but not used as a comparison in the
experimental section, in particular since they appear actually to
perform better than SWA in an early result of Fig.1. Note, again, that
the accuracy *with* variation should be looked at, since the attainable accuracy *without*
variation is completely irrelevant in this context.

**Questions:**

* The hardware results are interesting, however, it would have been
nice to compare also against the other training methods using the hardware
data. Since it is only tested against SGD, it is not clear how
e.g. 100% HSV, SWA, or the other methods would perform and thus it is hard
to judge the usefulness of WVAT for this particular data.

* I don't find where TTV is defined. It might be added Gaussian noise
during testing. While this is useful, it would be also interesting to
compare the robustness results using the standardized analog inference
model proposed by Rasch et al.. This would not only be a more
quantitative benchmark comparison against the state-of-the-art, it
also would measure robustness against more realistic noise sources
seen for in-memory analog inference hardware (beyond merely the
weight-related variations).